

# A game theoretic power control and spectrum sharing approach using cost dominance in cognitive radio networks

Sundus Naseer[1], Qurratul-Ain Minhas[1], Khalid Saleem[2],
Ghazanfar Farooq Siddiqui[2], Naeem Bhatti[1] and Hasan Mahmood[1]

[1] Department of Electronics, Quaid-i-Azam University, Islamabad, Pakistan
[2] Department of Computer Sciences, Quaid-i-Azam University, Islamabad, Pakistan

## ABSTRACT

The wireless networks face challenges in efficient utilization of bandwidth due to paucity of resources and lack of central management, which may result in undesired congestion. The cognitive radio (CR) paradigm can bring efficiency, better utilization of bandwidth, and appropriate management of limited resources. While the CR paradigm is an attractive choice, the CRs selfishly compete to acquire and utilize available bandwidth that may ultimately result in inappropriate power levels, causing degradation in network's Quality of Service (QoS). A cooperative game theoretic approach can ease the problem of spectrum sharing and power utilization in a hostile and selfish environment. We focus on the challenge of congestion control that results in inadequate and uncontrolled access of channels and utilization of resources. The Nash equilibrium (NE) of a cooperative congestion game is examined by considering the cost basis, which is embedded in the utility function. The proposed algorithm inhibits the utility, which leads to the decrease in aggregate cost and global function maximization. The cost dominance is a pivotal agent for cooperation in CRs that results in efficient power allocation. Simulation results show reduction in power utilization due to improved management in cognitive radio resource allocation.

## INTRODUCTION

In this modern era of high-speed communications, users and designers face the challenge of efficient spectrum utilization primarily due to its scarcity. In general, the usage of wireless radio spectrum is governed by allocating licenses to the primary users (PUs). In many scenarios, the allocated wireless bands are not fully used, which provides an opportunity to further improve spectral utilization. While the cognitive radio paradigm may have eased the problem of spectrum utilization, their deployment brings forth certain critical issues in radio resource management. In cognitive radio networks, if certain conditions are met, the PUs can opportunistically share their allocated bandwidth with secondary users (SUs) or unlicensed users. When the secondary users sense spectrum holes to transmit their information, competition begins with the peers in the network to use the resources. This induces antagonism among the SUs, which sometimes results in a

Corresponding author
Hasan Mahmood,
hasan@qau.edu.pk

hostile environment that seriously hampers the efficient utilization of the spectrum. Since there is no central coordinator in cognitive radio networks, all SUs selfishly try to maximize their throughput. Assuming the restriction of each node having a single radio transceiver, according to *Law, Huang & Liu (2012)* only one channel can be accessed by each SU in the network. As the nodes are making decisions independently, every SU aims for the best possible channel that motivates them to switch to lucrative channels. The SUs start to behave like sheep, which results in frequent switching of channels and causes rapid change in signal to interference and noise ratio (SINR) levels for all the users. The throughput of a SU depends upon the number of SUs sharing a spectrum with desired power levels that ultimately results in congestion. The power control algorithms are very effective in controlling interference and throughput, but in the absence of a central authority it becomes challenging and difficult. In addition, successive waveform adaptation mechanisms in a code division multiple access (CDMA) network can be employed to maintain the signal to interference ratio (SIR) threshold, which results in better SIR levels and improved resource sharing. Cognitive radios, therefore, need to collectively manage both the benefits and costs of channel switching and spectrum sharing along with energy utilization. A cognitive radio network is a specialized technology that provides opportunities for spectrum sharing in competitive environments that compel the users in a game theoretic environment to coordinate, which assists in mitigating the effects of conflicts in spectrum sharing. In order to improve the performance of a cognitive radio network, it is imperative to induce mechanisms that alleviate the adverse effects caused by uncontrolled interference.

The focus of this paper is on cooperative spectrum sharing that considers the amount of interference each node faces due to neighboring nodes while the cooperation problem is managed with game theoretic approach. The cooperative congestion game is proved to be a useful tool in resolving the adverse effects created by the selfish behavior of CRs within the network. The congestion game helps the CRs to make better decisions for network stability. These decisions, when analyzed in a Nash equilibrium perspective, proved to be cost effective for CRs. A unique cost function is deduced based on SIR and channel switching cost. The inverse signal to interference ratio game results in better response of the CRs that converge quickly to a unique Nash equilibrium. The power levels are adjusted by keeping the derivative of utility function to zero. The proposed inverse algorithm converges quickly and helps to accommodate a greater number of users within the network. This method is useful in mitigating congestion within the available bandwidths.

The proposed work can be applied to many practical systems that are based on cognitive radios. The application to cognitive radio wireless sensor network (CR-WSN) is interesting. The performance in power allocation and spectrum management can be enhanced by using the presented congestion game model. The proposed algorithm can also be used in emergency CR networks and public safety communications that use white space. Further applications include portable cognitive emergency network, medical body area networks (MBAN), and vehicular networks in reference to *Xiao et al. (2018)*.

## RELATED WORK

In pursuit to acquire the best channel in the wireless spectrum, the CRs are geared with hardware that assists to quickly switch between the channels. The primary reason to seek and switch to another channel is to mitigate the effects of interference as mentioned in *Southwell, Huang & Liu (2012)*. Several parameters contribute to the interference in a wireless network, which include transmission power and the choice of signature waveforms. In order to reduce interference, many adaptation algorithms are employed that use suitable waveforms on the channel opted by a SU. According to *Ulukus & Yates (2001a)* the greedy interference avoidance (IA) algorithms adapt the waveform codes sequentially. An iterative methodology to manage orthogonal sequences is proposed in *Anigstein & Anantharam (2003)*. The minimum mean squared error (MMSE) algorithms depend upon the stochastic receiver measurements; hence the convergence must be examined. Welch Bound Equality is achieved in *Anigstein & Anantharam (2003)* by distributed algorithms but convergence to optimal sequence set cannot be assured. The stability of eigen-iterative interference aware (IA) technique due to the addition and deletion of nodes in CDMA system is discussed in *Rose, Ulukus & Yates (2002)*. However, the convergence-speed experiment is not performed for greedy IA algorithms. Only a single receiver is assumed for the experiments as multiple receivers showed unstable behavior. The IA techniques, dealing with the distributed signature sequences, are further examined and discussed by *Popescu & Rose (2003)*, *Sung & Leung (2003)*, *Popescu & Rose (2004)*, and *Ulukus & Yates (2001b)* for multiple CR receivers and adaptations in asynchronous CDMA systems.

In wireless communication systems, power control is applied to compensate for fast fading, time-varying channel characteristics and to minimize battery power consumption especially in CDMA systems. Most of the power control algorithms are focused on a QoS of CRs. However, the nature of CRs show the dependency of power allocation decision on interference levels each CR receiver faces. Interference temperature as a critical decision maker within cognitive radio network is introduced in *Haykin (2005)*. The evolutionary issue discussed is the trust factor of other users that interfere with cognitive radios within the network. The separable game-theoretic framework for distributed power and sequence control in CDMA systems is modeled in *Sung, Shum & Leung (2006)*. It is established that if the equilibrium of the sequence control sub-game exists, only then the equilibrium of the joint control game exists. Hence, the convergence of joint control game is an open problem to be solved. The game theoretic analysis of wireless ad hoc networks is discussed in *Srivastava et al. (2005)*, and potential games have attracted large audience due to the existence of at least one Nash equilibrium. The congestion game model approach for resource allocation is discussed in *Ibrahim, Khawam & Tohme (2010)* and *Xu et al. (2012)*. The radio access interfaces (RAI) selection process is proposed as a non-cooperative congestion game model, where users share common set of resources. However, the practical implementation of RAI policy is tricky as the exact cost of the mobile users is not incurred to make their migration decisions. The users have the measured-based cost estimation. Similarly, the simultaneous migrations

proved to be damaging for the process and the pure Nash equilibrium is not assured. A local congestion game is formulated in *Xu et al. (2012)* that is proved to be an exact potential game. Spatial best response dynamic (SBRD) is proposed to achieve Nash equilibrium based on local information. The potential function reflects the collision levels within the network and can converge at any Nash equilibrium point, global or local. Thus, the Nash equilibrium leads to sub-optimal network throughput, the optimal Nash equilibrium point remains a challenging task to achieve. Different utility functions have been proposed in cognitive radio networks in recent research. The utility function based on ratio between user throughput and transmission power is proposed in *Saraydar, Mandayam & Goodman (2002)* along with the linear pricing terms. The QoS is analytically settled as utility in non-cooperative power control game. The Nash equilibrium achieved is not efficient. However, the existence of Nash equilibrium is not necessarily unique.

The proposed algorithm in *Kim (2011)* adapts its transmission power levels to the constant change in network environment to control the co-channel interference. Whereas, the convergence to the real-world scenarios is yet to be analyzed. A distributed power control through reinforcement learning is proposed in *Zhou, Chang & Copeland (2011)* that requires no information of channel interference and power strategy among users. The recent work on cognitive radio resource allocation is mostly based on non-cooperative game theoretic frameworks. The chaos-based game is formulated in *Al Talabani, Nallanathan & Nguyen (2014)* whose cost function is dependent on power vector and SINR values. The chaotic variable is a trade off between power and SINR. The power consumption of the proposed algorithm is less than traditional algorithms at the expense of 1–3 percent drift from average SINR. However, the affect of interference on primary user still needs to be studied.

A new SIR-based cost function for game theoretic power control algorithm is proposed in *Alpcan et al. (2002)*. The cost function is dependent on SIR logarithmically and is linear in power. However, the existence of the unique Nash equilibrium is only for a subset of the total number of active users. Another cost function based on weighted sum of linear power and SIR squared error is proposed in *Koskie & Gajic (2005)*. The static Nash equilibrium is achieved with low individual power level by compromising on SIR values. Similarly, in new findings the spectrum sharing is investigated in *Xiao et al. (2018)* for moving vehicles in heterogeneous vehicular networks (HVNs) consisting of the macrocells and the roadside units (RSUs) with cognitive radio (CR) technology. The non-cooperative game theoretical strategy selection algorithm is designed based on regret matching. The utility function is dependent on number of RSUs within the network, time of vehicle's presence (taken as constant) and number of vehicles present within the range of a RSU. The focus of the investigation is on correlated equilibrium that includes the set of Nash equilibrium. However, in the study the handover issues have not been considered when a vehicle user moves across the coverage of RSUs, which may cause failure for vehicle to RSUs connection, especially in high dynamic vehicular scenarios.

A chance-constraint power control in CR network is proposed by the Bernstein approach in *Zhao et al. (2019)* where channel gains are uncertain. The sum utility is attempted to maximize with outage probability constraints of CRs and PUs. In order to

achieve a convex problem, a protection function is formulated. Because of the uncertainty, the sum utility is reduced as compared to other solutions. The design specifications of optimal strategy space including power, speed and network information is introduced in *Mohammed et al. (2019)*. A non-cooperative game is formulated for this purpose. An energy efficient power control algorithm is proposed in *Zhou, Zhao & Yin (2019)*, in which protection margins for SINR and time-varying interference threshold are introduced. However, channel gain disturbances can be seen in simulation results. Intelligent reflecting surface (IRS) is a new technology applied in cognitive radios for interference minimization presented in *Zhou, Zhao & Yin (2019)*. An IRS is employed in proposed system to assist SUs for data transmission in the multiple-input multiple-output (MIMO) CR system. The performance gain can be increased by increasing number of IRS's phase shifts or deploying them at optimal locations. However, IRS technology needs an extensive study in the game theoretic perspective.

The congestion games are briefly discussed in the "Congestion Games". The rest of the paper consists of the system model and the proposed game model that leads to the Nash equilibrium. The proposed game algorithm and simulation results are discussed in the following sections along with the comparison to an exact potential game. The existence of Nash equilibrium is also proved. The conclusion and future perspectives to this work are also presented.

## CONGESTION GAMES

The congestion game is a useful tool in game theory when it comes to resource sharing. *Rosenthal (1973)* proposed the congestion game model in game theory for the first time, which was then followed by *Monderer & Shapley (1996)*. Monderer and Shapley proved that every congestion game is an isomorphic to an exact potential game. The payoff function of each user in a congestion game depends on the choice of resources it makes and the number of users sharing that resource. The payoff function of an exact potential game can be modeled as cost or latency function in a congestion game. The cost function induces a negative effect to congestion. This effect dominates with an increase in the number of players sharing the same resource. Furthermore, by establishing a global function, pure Nash equilibrium can be achieved. Congestion games can be defined as a tuple $(I, R, (S_i)_{i \in I}, (U_r)_{r \in R})$ where $I = 1, 2, \ldots, N$ denotes the set of players $N$, $R$ is the finite set of available resources, $(S_i)_{i \in I}$ is the strategy set of each player, $i \in I$ such that $S_i$ is the subset of the $R$ and $(U_r)_{r \in R}$ is the payoff function associated to the resources players opt as their strategies. The payoff function depends on the total number of players sharing the same resource. In general, players in congestion games aim to maximize their payoff function or minimize the total cost to achieve the Nash equilibrium.

### Cooperative congestion games

Congestion games deal with both the cooperative and non-cooperative players. In game theory, the cooperative games focus on the joint actions that players make and the resultant collective payoffs. The congestion externalities or the cooperative factors are involved in the process that may result in non-optimal equilibrium. However, according to

*Milchtaich (2004)* the equilibrium could be socially optimal, regardless of the fixed parameters affecting utilities, if the cost increases with increasing number of players. Thus, the need for cooperation is evident for optimal sharing of resources. In this work, the utility of each player decreases as the size of players set sharing the same resource increases. The players are heterogeneous as they achieve different payoffs by opting for the same choice of resource.

In this paper, we propose that socially optimal Nash equilibrium of cooperative congestion game is achieved by the negative utility function considered as the cost paid by each user. This leads to the ultimate minimization of aggregate cost interpreted as the global function maximization.

## THE SYSTEM MODEL

The CR network consists of multiple transmit and receive node pairs. The SINR of each node in CDMA system is dependent on the correlation with the waveforms of other users sharing the same spectrum; their transmit power levels and spectrum characteristics. Waveforms of nodes are represented by the signal space characteristics that show nearly orthogonal signal dimensions (either in frequency, time, or spreading waveforms), as mentioned by *Anigstein & Anantharam (2003)*. One of the important aspects that we consider in this paper is reduction in inverse signal to interference ratio (ISIR) by using efficient spectrum sharing based on the correlation between waveforms of users sharing that spectrum along with power optimization. Pseudo random sequences are taken as they hold various properties of white noise with minimum auto and cross-correlation. The base data pulses directly multiply with the pseudo random sequences and each resultant waveform pulse represents a chip. The resultant waveform signals are non-overlapping rectangular pulses of amplitude +1 and −1, *Rappaport (1996)*. Consider a network that consists of $N$ cognitive radios, which are distributed randomly in the deployment area. The $K$ transmission frequency bands are available in the network, where $K < N$.

The spectrum sharing of the *CRs* is modeled as a normal form of congestion game, $G = (N, \{S\}, U_{i \in N})$. The strategy space of users is $S = (S_1 \times S_2 \times \cdots \times S_N)$. Here $S_{i \in N}$ is for the set of player $i$ that consists of two subsets, $ch_i = \{ch_1, ch_2, \ldots, ch_k\}$ the set of available channels within the network and $S_i = \{s_1, s_2, \ldots s_N\}$, set of signature sequences $\forall\ i \in \{1,2,\ldots,N\}$. The utility function of player $i$ is expressed as $U_{i \in N}$ that also includes the cost function.

In the proposed game model, the ISIR can be expressed as:

$$\gamma_i = \frac{s_i^H s_j s_j^H s_i p_j h_{ji}^2}{p_i h_{ii}^2} \tag{1}$$

where, $s_i$, $s_j$ are the signature sequences of the nodes and $s_i^H$, $s_j^H$ are the transpose of these sequences. $p_i$ is the transmit power of node $i$ that is adaptable at each iteration. The link gain of the nodes in the networks is represented as $h_{ij}$. However, the gain remains constant as the network topology is fixed for simplicity. The model can easily be applied on dynamic network to make it more practical. Random waypoint mobility model is suitable for this purpose. If CRs compete for unlicensed bands then their fully cooperative

behavior is considered. This cooperation helps them to maintain the stable network conditions even in a dynamic network. The dynamic topology helps in motivating the CRs for cooperation to achieve better utility as given by *Wang, Wu & Liu (2010)*. Hence the proposed model in dynamic conditions can work in an efficient manner, however, it takes more time to get the desired results.

The utility of a player comprises of the benefit of minimum correlation with other players sharing the same channel and the cost of choosing that channel.

$$U_i(s_i, s_{-i}) = B_i(s_i, s_{-i}) - C_i(s_i, s_{-i}) \tag{2}$$

where, $s_{-i}$ is the strategy set of all the players except player $i$ that can be denoted as: $s_{-i} = (s_1 \times s_2 \times \ldots s_{i-1} \times s_{i+1} \ldots \times s_N)$. Here, $B_i$ defines the benefit user attains for a particular choice of strategy and $C_i$ is the cost.

## INVERSE UTILITY CONGESTION GAME

The purpose of the spectrum sharing congestion game is to reach a suitable utility level at which network achieves Nash Equilibria. The utility function might not be maximized as the cost of spectrum sharing and adaptation of a new suitable channel is involved. The cost function of a player $i$ is:

$$C_i(s_i, s_{-i}) = \left( \frac{b_p \gamma_i(s_i, s_{-i}) + c_s}{p_i} \right) \tag{3}$$

where, $b_p$ is the battery power of node $i$ transmitter, $\gamma_i(s_i, s_{-i})$ is the inverse signal to interference ratio of player $i$ at some particular channel, and $c_s$ is the channel switching cost. The channel switching cost increases as the player keeps on shifting from one channel to another in search of the optimal result. Hence, the cumulative switching cost can be defined as:

$$C_{switch} = \begin{cases} +1, & \text{if } ch_{i, iter+1} \neq ch_{i, iter} \\ 0, & \text{otherwise} \end{cases} \tag{4}$$

The cost increases by a factor of 1 every time a player switches its strategy from one channel to another. But if the channel remains the same at next iteration the switching cost becomes 0. Since the benefit of choosing a channel and sharing it with other users is in terms of the minimum cross-correlation that is

$$B_i(s_i, s_{-i}) = -\left( \frac{s_i^H(X)s_i}{p_i} \right) \tag{5}$$

The sequential congestion game is played iteratively. The users make their choices after analyzing the interference faced at their particular channel, and on other channels. The interference faced by each user is dependent on the correlation and the transmitting power of users sharing same channel. The waveforms of users sharing same channel are replaced by the eigenvector corresponding to the smallest eigen values of correlation matrix $X$, as narrated by *Menon et al. (2005)*. The iterative game helps in reaching the minimum correlation set of each player that increases the benefit function by reducing

interference at each channel. The utility function of the proposed game model is derived after substituting (3) and (5) in (2) as:

$$U_i(s_i, s_{-i}) = \left(-\frac{s_i^H(X)s_i}{p_i}\right) - \left(\frac{b_p \gamma_i(s_i, s_{-i}) + c_s}{p_i}\right) \tag{6}$$

$$U_i(s_i, s_{-i}) = \left(-\frac{s_i^H(X)s_i}{p_i}\right) - \left(b_p\left(\frac{s_i^H s_j s_j^H s_i p_j h_{ji}^2}{p_i^2 h_{ii}^2}\right)\right) - \frac{c_s}{p_i} \tag{7}$$

The users occupy the channel where they receive maximum utility. Whereas, the cost paid in terms of consumed power motivates the users to cooperate and opt a suitable band option for the network. The users then optimize their power according to the current channel conditions.

The game can be extended to simultaneous iterative play. However, users need to maintain the history chart in order to realize the behavior of other players. The history of opponents is helpful in making better decisions in the next iteration.

### Effect of cost on utility function

The utility function is specified in (2) of the proposed game model. As discussed above, each user measures its benefit and cost at each channel sequentially and then opt for the channel with maximum utility. The utility function of each user keeps on increasing as the game starts, since the CRs are selfish and opt for their best possible resource. After certain iterations, the cost factor starts to dominate players' benefit. The channel switching cost increments after every single change. The cost dominance is considered as an influencing factor for CRs to cooperate, resulting in a cooperative congestion game.

The dominance of cost function enforces the CR users to reach for the suitable utilities. This leads to the negative utility function values; the cost starts to increase and benefit decreases. However, at a certain point of equilibrium the players reach their best possible spectrum choice with minimum interference. The negative resultant utility function models the inverse utility congestion game.

### Inverse power control algorithm

Most of the power-control algorithms are focused on cellular networks where satisfying the *QoS* constraint is a stringent requirement. In CR networks, transmitters increase power to cope with channel impairments and increasing levels of interference in an inconsiderate and competitive manner. Within the spectrum sharing framework, a network strongly opposes the secondary users to transmit with arbitrarily high power and interferes with the *QoS* of the primary users. Hence, to achieve the target SINR, $\sigma_i$, secondary user power is required to be kept low. In order to find out the change in utility with respect to power of a node, we take derivative of (7),

$$\frac{dU_i(s_i, s_{-i})}{dp_i} = \left(\frac{s_i^H(X)s_i}{P_i^2}\right) + 2b_p\left(\frac{s_i^H s_j s_j^H s_i p_j h_{ji}^2}{p_i^3 h_{ii}^2}\right) + \frac{c_s}{p_i^2} \tag{8}$$

Putting the derivative equal to zero,

$$\frac{dU_i(s_i, s_{-i})}{dp_i} = 0 \tag{9}$$

$$\left(\frac{s_i^H(X)s_i}{p_i^2}\right) + 2b_p \left(\frac{s_i^H s_j s_j^H s_i p_j h_{ji}^2}{p_i^3 h_{ii}^2}\right) + \frac{c_s}{p_i^2} = 0 \tag{10}$$

$$\frac{1}{p_i^2}(s_i^H(X)s_i) + 2b_p \left(\left(\frac{s_i^H s_j s_j^H s_i p_j h_{ji}^2}{p_i h_{ii}^2}\right) + c_s\right) = 0 \tag{11}$$

$$(s_i^H(X)s_i) + c_s = -2b_p \left(\frac{s_i^H s_j s_j^H s_i p_j h_{ji}^2}{p_i h_{ii}^2}\right) \tag{12}$$

$$p_i h_{ii}^2(s_i^H(X)s_i) + c_s = -2b_p s_i^H s_j s_j^H s_i p_j h_{ji}^2 \tag{13}$$

$$p_i = \left(\frac{-2b_p s_i^H s_j s_j^H s_i p_j h_{ji}^2}{h_{ii}^2(s_i^H(X)s_i + c_s)}\right) \tag{14}$$

Assuming orthogonal signature sequences $(s_i^H(X)s_i)$ reaches to 1, the change in total power, $p_i$, of a user $i$ at time $t$ is:

$$p_{i_{t+1}} = \left(\frac{-2b_p p_j h_{ii}^2}{h_{ii}^2(1 + c_s)}\right) \tag{15}$$

We know that $SIR_i$ reaches to maximum at minimum correlation between sequences of the users. Ideally, the signature sequences are considered orthogonal to each other, which means $(s_i^H s_j s_j^H s_i) = 1$. Therefore,

$$SIR_{i, max} = \sigma_i = \frac{p_i h_{ii}^2}{p_j h_{ii}^2} \tag{16}$$

The updated power at each iteration can be calculated as:

$$P_{i_{t+1}} = \left(\frac{SIR_{i, max}}{SIR_i}\right) P_i \tag{17}$$

$$P_{i_{t+1}} = \sigma_i \gamma_i P_i \tag{18}$$

After substituting (1), (15) and (16) in (18), we get

$$P_{i_{t+1}} = \left(\frac{-2b_p s_i^H s_j s_j^H s_i p_j h_{ji}^2}{h_{ii}^2(1 + c_s)}\right) \tag{19}$$

Hence, the suitable power of each user at each iteration is calculated by $P_{i_{t+1}}$.

## Global function

The utility of each user influences the strategy set $s_{-i}$ of opponents in games. This influence is made to minimize the interference within the network. The impact of utility of each user is projected in the form of a global function. In the proposed game framework, global function is the negative sum utilities of each user.

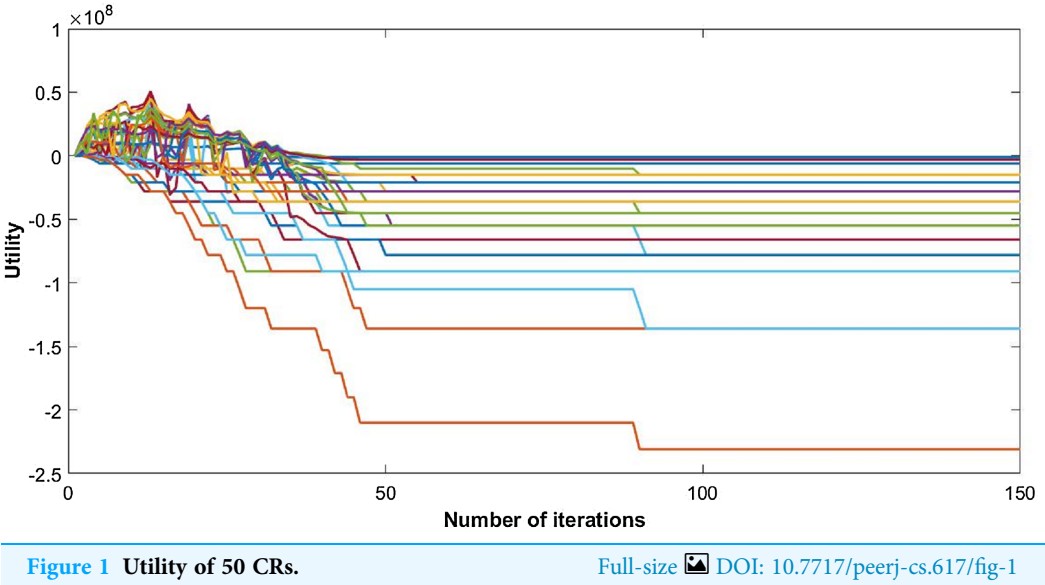

**Figure 1  Utility of 50 CRs.**                

$$P_s = - \sum_{i=1,i\neq j}^{N} \left[ \max_{s_i \in S} U_i(s_i, s_{-i}) \right] \qquad (20)$$

The convergence of global function shows social stability in the CR network, where each user is operating at its best possible channel without creating any hindrance in transmission of other users or even PU (in case of underlay systems).

## SIMULATIONS RESULTS

The inverse utility congestion game is evaluated using a variable number of CRs. The network considered is of area 50 × 50 m with 50 nodes, uniformly distributed to share $K = 4$ available channels. After iteratively playing the game, the utilities of players with three signal space dimensions are shown in Fig. 1.

The utility of nodes starts to increase as each node acts selfishly and tries to maximize its own payoff. This selfish behavior increases the cost of nodes and ultimately leads to the negative utility. Users keep on changing their channels until they reach an optimum point of utility. Convergence of channel allocation process is shown in Fig. 2. It is observed that users select their suitable channels after a few iterations, but the signature sequences take time to converge. The correlation between waveforms reach its minimum level with delay; since, the change is so small that it hardly influences the channel decision made by each user. However, it ultimately converges to its optimal point as shown in Fig. 2.

The convergence of the global function of the network is shown in Fig. 3. The equilibrium point at which users within the network coordinate with each other and the network becomes stable is the maxima of the global function. The stability ensures the existence of a unique Nash equilibrium. Here the noticeable trend is the switching of convergence point twice at near 100*th* iteration. From game theoretic perspective, it could be the shifting of local maxima to global one. However, trends need further study.

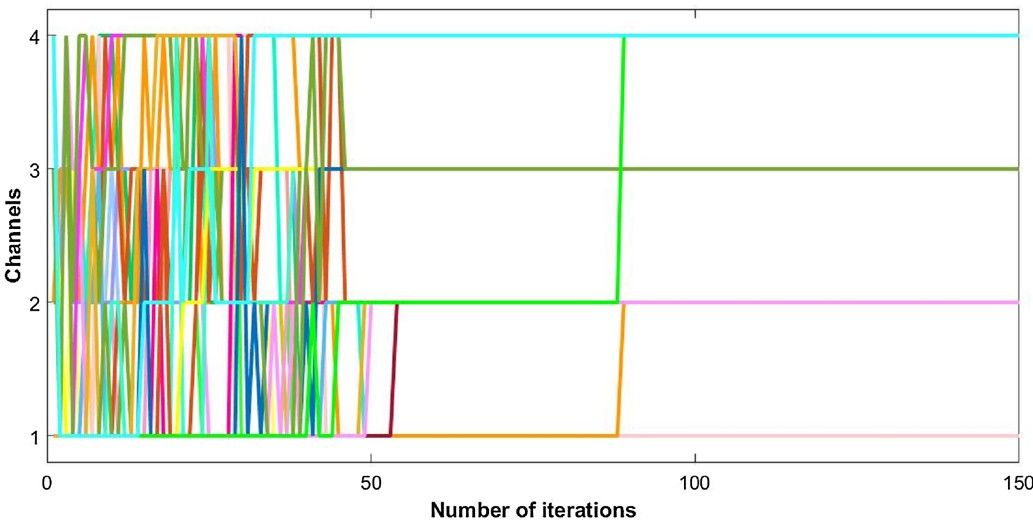

**Figure 2 Convergence of CR channel allocation process with 50 users.**

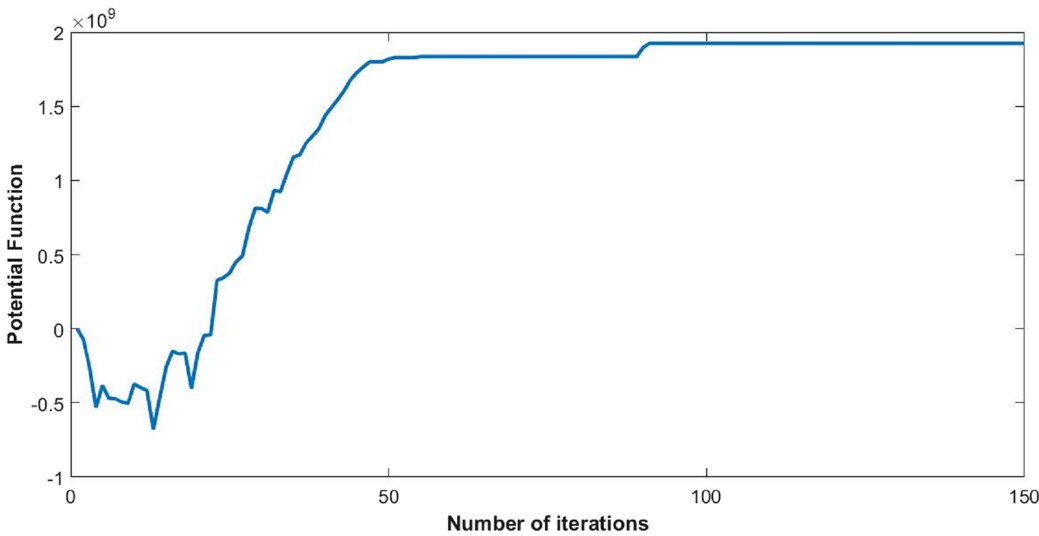

**Figure 3 Convergence of global function ($P_s$) with 50 users.**

According to *Zhou, Zhao & Yin (2019)* the global function of the congestion game is isomorphic to the potential function of the exact potential games.

The steady state of the function is pure Nash equilibrium, where no further change in strategy of a user can be beneficial. In other words, no unilateral deviation from the equilibrium point gives incentive to any user. Global function shows the overall throughput of the network. The reduction in ISIR within the network is the inverse of potential function that can be seen in Fig. 4. This reduction depicts the minimum interference within the network and optimum power levels of each CR. Since they agree to cooperate, transmission power of each user starts reducing, as shown in Fig. 5.

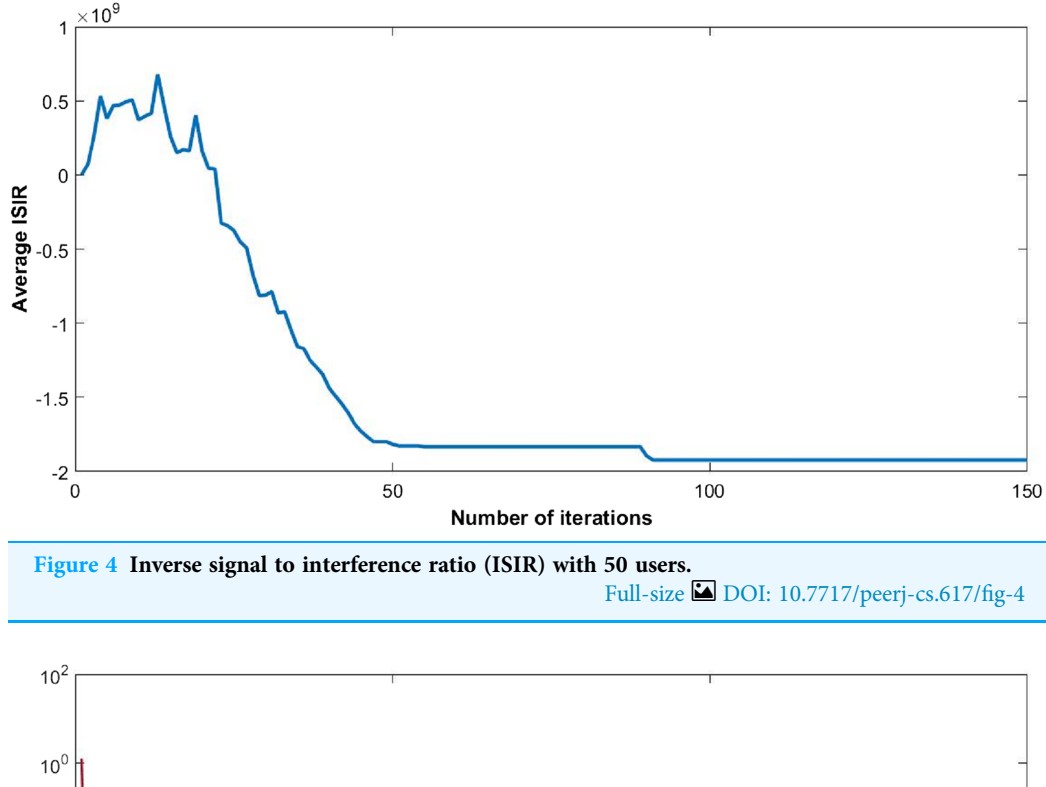

**Figure 4** Inverse signal to interference ratio (ISIR) with 50 users.

**Figure 5** Power of 50 CRs at each iteration.

The behavior of the cognitive nodes is more clarified in Figs. 6 and 7. The pattern of five users show how they switch between channels and opt the suitable utility.

A quick comparison of convergence is shown in Table 1 with a chaotic optimization method of power control given by *Al Talabani, Nallanathan & Nguyen (2014)* and a non cooperative game theoretic approach in heterogeneous vehicular networks (HVNs) with correlated equilibrium presented by *Xiao et al. (2018)*

The proposed inverse power control (IPC) algorithm is noticeably accommodating more number of users with minimum interference than other algorithms in literature. This prevents the wastage of bandwidth of licensed band when it is free to use for CR. Although power of each user slightly decreases as the cost is dominant, this helps in

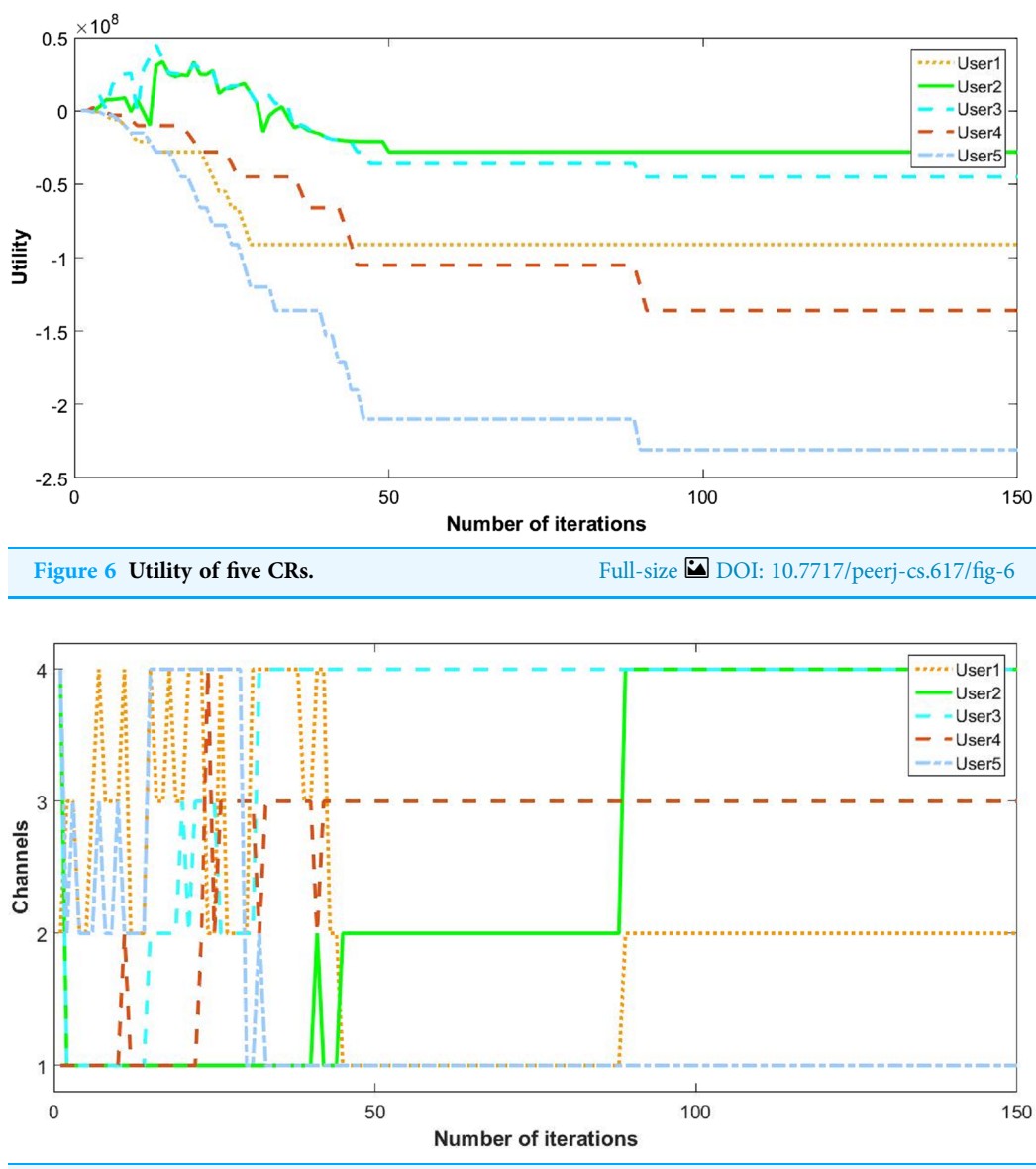

Figure 6 Utility of five CRs.             

Figure 7 Convergence of CR channel allocation process of five users.

Table 1 Convergence of algorithms.

|  | Accommodated users | Convergence point | Convergence unit |
|---|---|---|---|
| Chaotic Optimization of Power Control | 5 | 50[th] Iteration | Power and Information Rate |
| Non-Cooperative HVN Game | 5 | 20[th] Iteration | Average Utility |
| Proposed IPC | 10 | 9[th] Iteration | Utility and Power |

minimizing interference between users. Also, the proposed algorithm is simpler as compared to the current state-of-the-art algorithms. Figures 8, 9 and 10 shows the convergence of the network with 10 CRs.

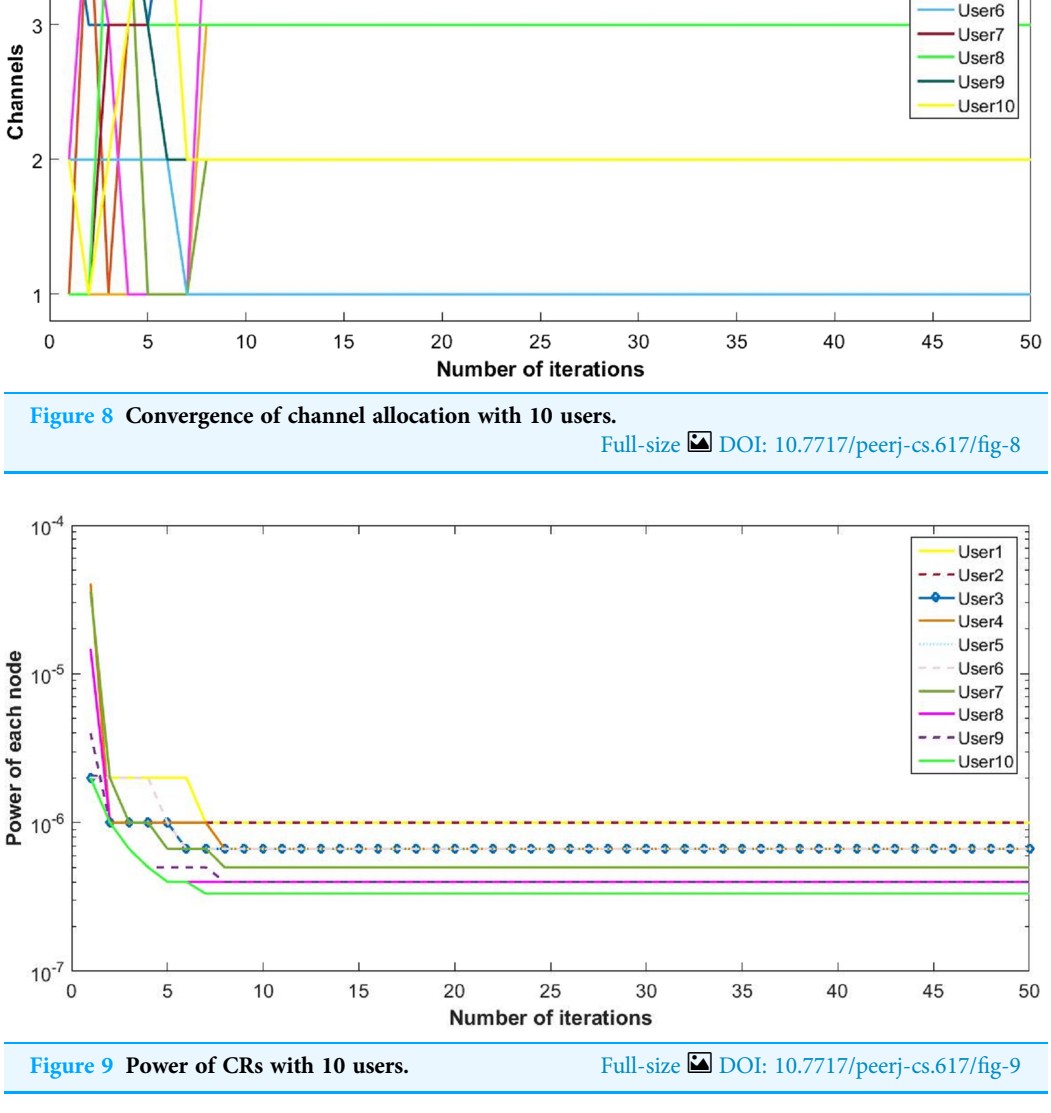

**Figure 8 Convergence of channel allocation with 10 users.**

**Figure 9 Power of CRs with 10 users.**

## Congestion game isomorphic to potential game

As described above, the congestion game is an isomorphic to a potential game. We observe the results of inverse utility game without cost that can be considered as a potential game model as shown in Figs. 11 and 12.

Similarly, the ISIR of the network remains unstable for a long period of time, since selfish nodes in CR network do not consider the cost initially. Later, nodes must bear cost in terms of a destabilized network and excessive interference at each channel that forces them towards cooperation. This process results in system delay and convergence is not always guaranteed. The Fig. 11 shows the unstable trend of ISIR until the convergence is achieved.

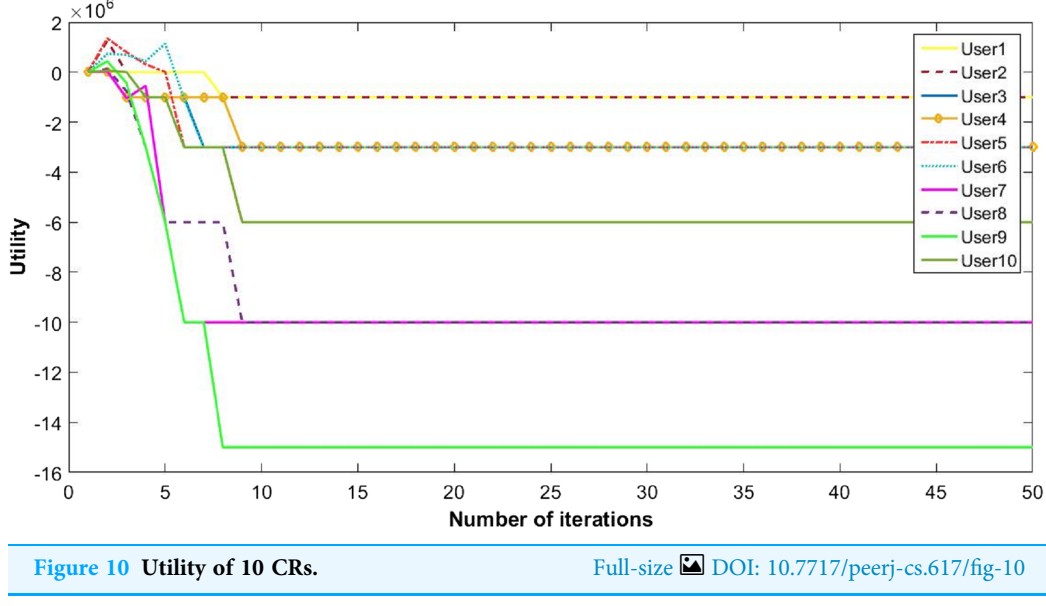

**Figure 10 Utility of 10 CRs.**               

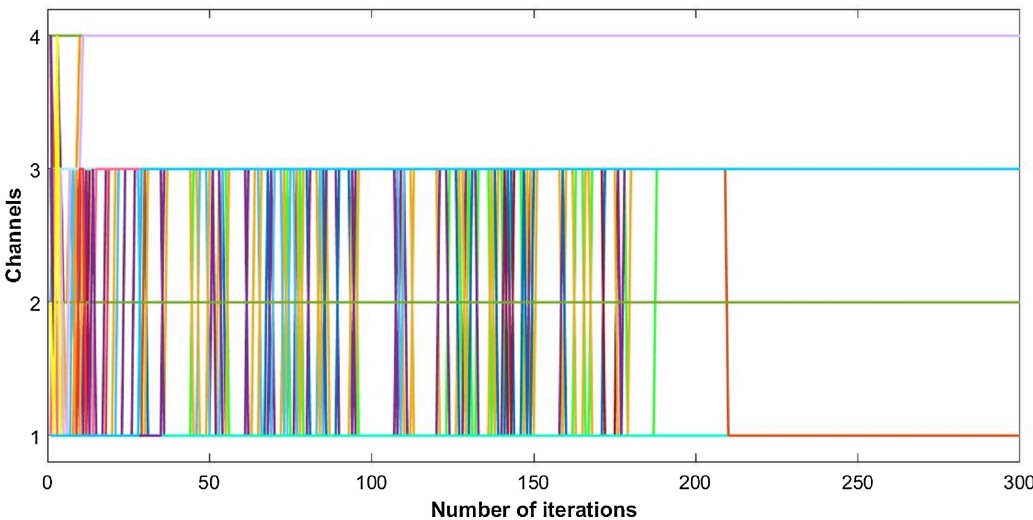

**Figure 11 Convergence of CR channel allocation process without cost factor.**

## Existence of Nash equilibrium

Players cannot improve further in terms of utility when they reach to the optimum point, which is called as the Nash equilibrium. The strategy set $s_i \in S$ is considered as Nash equilibrium if

$$U_i(s_i) \geq U_i(s_i^*, s_i) \forall i \in N, s_i \in S \tag{21}$$

Since a combination of potential (coordination) games and dummy games together make congestion game $G = (N, \{S\}, \{U_{i \in N}\})$, where all users of the game choose their responses in rational round robin order until no better solutions are selected to achieve Nash equilibrium. For this reason, best response dynamics of a coordination game can be

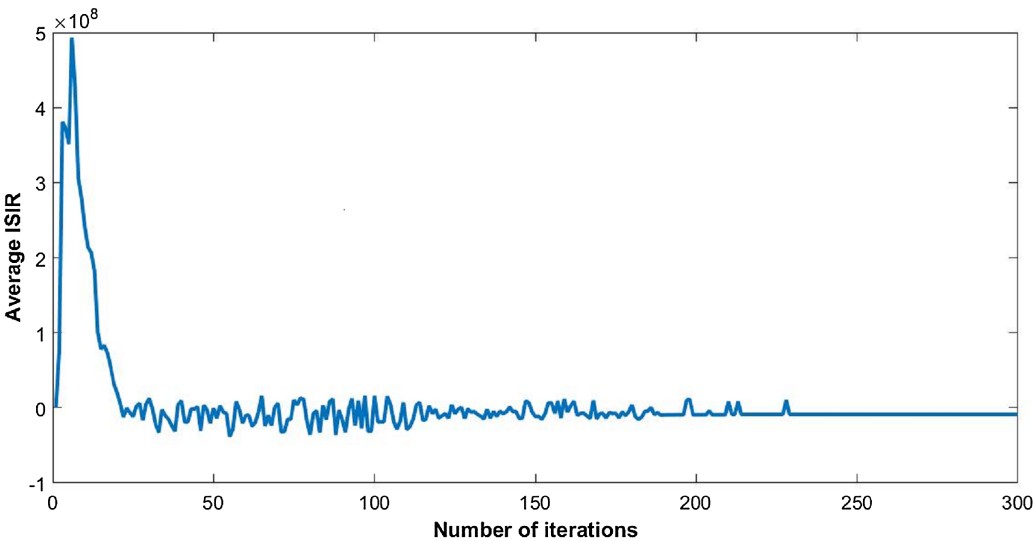

**Figure 12 Inverse signal to interference ratio in CR network without cost factor.**

placed equal to its corresponding potential/congestion game. Existence of unique Nash equilibrium is assured in proposed game due to its continuous, bounded global function and one global maximum that is a unique characteristic of Nash equilibrium, which is maxima of global function.

The round-robin best response iteration, $\phi$, is the convergence point and is known as eigen-iteration. On the other hand, despite being convergent strategy that is the set of eigenvectors, $R_{rr}$ (set of vectors corresponding to minimum eigen values of the correlation matrix), it might not be considered as most advantageous tactic and can be clarified through the following lemma presented in *Hicks et al. (2004)*.

**Lemma 1** *Let $s_i \in S \ \forall i \in N$, S and $\phi$ be the best response dynamic. If $s \in \phi \ \forall \ i \in 1,2,...,N$, $s_i$ is an eigenvector of $R_{rr} = sps^T + R_{zz}$.*

Here, $R_{zz}$ is the covariance matrix of additive gaussian noise, which is ideally assumed to be 0. In order to satisfy Lemma 1, fixed points under eigen iteration in a sequence are arranged in proposed algorithm. Also, to analyze convergence in potential and corresponding congestion games, some of the conditions of Zangwill's convergence theorems in *Dubois (1973)* are essential. The mentioned conditions are:

1. At every iteration all of the strategy points, $s_i$ should be present in a compact set, $s_i \subset S$.
2. There is a continuous set: $P : S \rightarrow \mathbb{R}$ such that

    a) If $s_i \in S$ is not a Nash equilibrium, in that case $P(s^\star) > P(s)$ for any $s^\star \in \phi(s)$.

    b) If $s \in S$, in that case either the algorithm terminates or for any $s^\star \in \phi(s)$, $P(s^\star) \geq P(s)$.

3. When any convergence point is considered, $s_i \rightarrow s'$, if $s'$ is not a Nash Equilibrium; in that case the convergence point is $P(s^\star)$; hence lim $P(s^\star)>P(s')$. Then $s^\star \in \phi(s_i)$ reaches to a Nash equilibrium.

Since in a single round-robin iteration all of the best responses in signature sequence space (strategy space) $s_N \in S \forall i \in N$ that can be represented by $\phi$ and $s_N$ is a compact set; the first condition of above theorem is fulfilled. Furthermore, by considering the definition of Nash equilibrium if $s^\star$ is the set of NE, then both above conditions $2(a)$ and $2(b)$ are fulfilled by using best response iteration of function $P(s)$. Let $\phi$ be an upper semi-continuous correspondence, a compact space having closed graph, then third condition of above theorem is fulfilled. Due to the reason that all conditions are fulfilled, it can be argued that under the best response dynamics the potential games as well as the congestion games converge to a Nash equilibrium.

Therefore, it can be concluded from the above discussion that the proposed model of inverse utility congestion game also shows evidence of best response convergence. Furthermore, due to this best response convergence, QoS is assured at better level and there is also an iterative decrease in interference but at the cost of maximum individual utility. Though, utilities of the users are best possible ones and are not maximum, still each CR user is getting better payoff at each iteration than the previous iterations, which at the end leads to an appropriate payoff for each user and becomes convergent point in the game. The utility function is the concave function of strategies and has a unique maximum. The convergence to a Nash equilibrium is present at the maximum of the global function, as discussed above, it is bounded and continuously differentiable. Users get the best possible set of waveforms and appropriate sub-channels at the minimum level of inverse signal to interference ratio function.

As far as the complexity of the algorithm is concerned, it is locally executed by every user, therefore, the process involves sensing and making decisions on channel selection, power allocation and choice of waveforms independently. This requires constant time at the user node to execute the algorithm, with complexity of O(1).

## CONCLUSION

In this paper, an inverse utility congestion game is proposed to mitigate interference and congestion in CR network. The game model is designed to formulate the power optimization problem through waveform adaptation process for the efficient utilization of available resources. An intelligible algorithm is applied for efficient spectrum sharing of the users that possibly takes less time in reaching Nash equilibrium. The target SIR is achieved through proposed algorithm; however, time lapse can be seen in the convergence of nearly orthogonal waveform correlation matrix. The utility is not maximum but at suitable level for each node.

In this paper, a comparison of the inverse utility congestion game with and without cost effect is also analyzed. The Nash equilibrium is also achieved in later scenario, but the system experiences delay in convergence. Hence, the cost dominance is helpful in attaining cooperation among nodes. This cooperation leads to congestion mitigation in scarce bandwidth resources that could be useful in multiple practical systems.

### Funding

The authors received no funding for this work.

### Competing Interests

The authors declare that they have no competing interests.

### Author Contributions

- Sundus Naseer conceived and designed the experiments, performed the experiments, analyzed the data, performed the computation work, prepared figures and/or tables, and approved the final draft.
- Qurratul-Ain Minhas conceived and designed the experiments, performed the experiments, analyzed the data, performed the computation work, prepared figures and/or tables, and approved the final draft.
- Khalid Saleem conceived and designed the experiments, performed the experiments, analyzed the data, performed the computation work, prepared figures and/or tables, contributed analysis tool, and approved the final draft.
- Ghazanfar Farooq Siddiqui conceived and designed the experiments, performed the experiments, analyzed the data, performed the computation work, prepared figures and/or tables, authored or reviewed drafts of the paper, contributed materials, and approved the final draft.
- Naeem Bhatti conceived and designed the experiments, performed the experiments, analyzed the data, performed the computation work, authored or reviewed drafts of the paper, and approved the final draft.
- Hasan Mahmood conceived and designed the experiments, performed the experiments, analyzed the data, performed the computation work, authored or reviewed drafts of the paper, and approved the final draft.

### Data Availability

The MATLAB code are available in the Supplemental Files.

### Supplemental Information

Supplemental information for this article can be found online at http://dx.doi.org/10.7717/peerj-cs.617#supplemental-information.

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
