# Peer review of "A game theoretic power control and spectrum sharing approach using cost dominance in cognitive radio networks"

_PeerJ Computer Science, doi:10.7717/peerj-cs.617_

## Round 0.1 · original submission · Major Revisions

The authors need to improve the writing of this paper. The literature review is not complete. Some notations are missing in this paper. The contributions of this paper should be clearly stated.

Reviewer 1 ·

Basic reporting

This paper has studied a game theoretic cost dominant approach for cognitive radio networks. In general, this paper is interesting and good but could be improved. Some typos should be corrected. For example, in line 218, there should be a space between 'benefit' and 'Wang. Line 266: 'Figure2'->'Figure 2'.The title and abstract may be rephrased. Related work section is very long and could be more concise.

Experimental design

Experimental design is OK but could be improved. Some figure could be made clearer, for example, Fig. 2, 5, 6 are difficult to read.

Validity of the findings

The power control in cognitive radio has been studied extensively in the past. The authors should state clear what the contribution their papers has brought compared to the existing state-of-the-art. The authors are also suggested to compare their findings with the existing solutions in the literature and show the improvement.
Also, it is better you can analyze more on the convergence and complexity theoretically of the algorithms.

Additional comments

It is better the authors can show how the proposed solution can be applied in the practical system. It seems just the theoretical analysis conducted in the current version.

Reviewer 2 ·

Basic reporting

English grammar and sentence structure need to be rechecked. There are many irregularities in writing, such as:
1) The first sentence in abstract, "In wireless networks, poor communication, and congestion results from an increase in
demand of already scares bandwidth resources."
2) The third sentence in abstract, "While the CR paradigm is an attractive choice, the CRs selfishly compete to
acquire and utilize available bandwidth that may ultimately results in power allocation,
causing degradation in Quality of Service of the network"
3) Page 5, line 34, "In cognitive radio networks the PU"---->“In cognitive radio networks, the PU”, line 41, what is "Law et al.."? The same problem appears on line 46.
4) Line 190, "In this paper we propose..."-->"In this paper, we propose..."

Experimental design

1) The reasearch question is defined. The numerical results and relevant code are provided. However, more clearly and prominently statements should be given on how reasearch is different from or outperforms the existing research works such as the papers "A. Al Talabani, A. Nallanathan and H. X. Nguyen, "A Novel Chaos Based Cost Function for Power Control of Cognitive Radio Networks," in IEEE Communications Letters, vol. 19, no. 4, pp. 657-660, April 2015, doi: 10.1109/LCOMM.2014.2385068", "Z. Xiao et al., "Spectrum Resource Sharing in Heterogeneous Vehicular Networks: A Noncooperative Game-Theoretic Approach With Correlated Equilibrium," in IEEE Transactions on Vehicular Technology, vol. 67, no. 10, pp. 9449-9458, Oct. 2018, doi: 10.1109/TVT.2018.2855683."

2) New technolgy is developped recently, for example the Intelligent Reflecting Surface (IRS), the CR throughput can be further improved with assistance of IRS. The authors should mention relevant research work as the latest research background, such as the paper "Lei Zhang et al., Intelligent Reflecting Surface Aided MIMO Cognitive Radio Systems, in IEEE Transactions on Vehicular Technology, vol. 69, no. 10, pp. 11445-11457, Oct. 2020, doi: 10.1109/TVT.2020.3011308".

Validity of the findings

no comment

Additional comments

A congestion game model is proposed in this paper to mitigate interference and congestion in CR network. However, some places that may confuse the reader require careful explanation by the author.
1) The result of the game is equivalent to minimize the inverse SIR, why not directly aim to maximize the SIR? It seems that they have a same Nash equilibrium.
2) In equations (6) and (7), the first terms after the equal signs, which is the benefit of minimum correlation with other players, should be positive (without "-") from the perspective of formula 3. Why do you add "-"?
3) Please explain the unstabel state when the number of iterations is 100 in Figure 2.
4) The first occurrence of mathematical symbols should be explained, such as "B_i()" and "C_i()" in equation (2).

---

## Round 0.2 · Minor Revisions

The reviewers still have some minor comments. Please update them accordingly. Please proofread this paper carefully.

Reviewer 1 ·

Basic reporting

The format of the paper should be improved.

The figures are not clear.

Introduction should be Section I?

In Simulation section, section 5, second paragraph, there is ''Figure. ??''

Experimental design

Simulation part could be improved. The figures are difficult to read.

In Simulation section, section 5, second paragraph, there is ''Figure. ??''

Validity of the findings

Please improve your simulation part. Some figures are not clear and it is difficult to read.

Additional comments

The paper should be future improved. For example, the figures are not clear. Introduction Section should be Section I? There are several typos in the paper.

Reviewer 2 ·

Basic reporting

My questions have been answered appropriately. The quality of the revised paper has been improved. However, the grammar and typos should be checked again through the whole article. For example, in the third sentence of the abstract, "....that may ultimately results in inappropriate...." should be "...that may ultimately result in inappropriate...".

Experimental design

no comment

Validity of the findings

no comment

---

## Round 0.3 · accepted · Accept

The authors have addressed all the comments.